# Immunological Markers, Prognostic Factors and Challenges Following Curative Treatments for Hepatocellular Carcinoma

**DOI:** 10.3390/ijms221910271

**Published:** 2021-09-24

**Authors:** Soon Kyu Lee, Sung Won Lee, Jeong Won Jang, Si Hyun Bae, Jong Young Choi, Seung Kew Yoon

**Affiliations:** 1Division of Hepatology, Department of Internal Medicine, College of Medicine, The Catholic University of Korea, Seoul 06591, Korea; blackiqq@gmail.com (S.K.L.); garden@catholic.ac.kr (J.W.J.); baesh@catholic.ac.kr (S.H.B.); jychoi@catholic.ac.kr (J.Y.C.); yoonsk@catholic.ac.kr (S.K.Y.); 2The Catholic University Liver Research Center, College of Medicine, The Catholic University of Korea, Seoul 06591, Korea

**Keywords:** hepatocellular carcinoma, recurrence, tumor microenvironment, resection, liver transplantation, ablation, outcomes, immune checkpoint blockers, immune suppression

## Abstract

Hepatocellular carcinoma (HCC) is one of the leading causes of cancer-related mortalities worldwide. Patients with early-stage HCC are eligible for curative treatments, such as surgical resection, liver transplantation (LT) and percutaneous ablation. Although curative treatments provide excellent long-term survival, almost 70–80% of patients experience HCC recurrence after curative treatments. Tumor-related factors, including tumor size, number and differentiation, and underlying liver disease, are well-known risk factors for recurrence following curative therapies. Moreover, the tumor microenvironment (TME) also plays a key role in the recurrence of HCC. Many immunosuppressive mechanisms, such as an increase in regulatory T cells and myeloid-derived suppressor cells with a decrease in cytotoxic T cells, are implicated in HCC recurrence. These suppressive TMEs are also modulated by several factors and pathways, including mammalian target of rapamycin signaling, vascular endothelial growth factor, programmed cell death protein 1 and its ligand 1. Based on these mechanisms and the promising results of immune checkpoint blockers (ICBs) in advanced HCC, there have been several ongoing adjuvant studies using a single or combination of ICB following curative treatments in HCC. In this review, we strive to provide biologic and immunological markers, prognostic factors, and challenges associated with clinical outcomes after curative treatments, including resection, LT and ablation.

## 1. Introduction

Hepatocellular carcinoma (HCC) is the most common primary liver cancer and the third leading cause of cancer-related mortality worldwide [1,2]. The major risk factors for developing HCC include hepatitis B virus and hepatitis C virus infection, heavy alcohol intake, nonalcoholic fatty liver disease, diabetes, and dietary toxins [2,3]. For high-risk patients, although detailed target populations are different, clinical guidelines recommend 6-month ultrasound surveillance to detect and treat HCC in the early stage [4,5,6,7].

Patients with early-stage HCC, commonly represented as Barcelona Clinic Liver Cancer (BCLC) stage 0 or A, are eligible for curative therapies, including surgical resection, liver transplantation (LT), and percutaneous ablation [6,8,9]. Although the five-year survival rates of these patients are higher than 70%, approximately two-thirds of patients treated with resection or percutaneous ablation experience HCC recurrence within 5 years [10,11,12,13]. Moreover, the risk of HCC recurrence persists even after 5 recurrence-free years after treatment with curative therapies [14].

Consequently, it is pivotal to identify high-risk patients for recurrence after curative treatments. Traditionally, tumor-related factors such as tumor size and number are known to be associated with recurrence and have been used to identify eligible patients for curative treatments [15,16]. Other histologic factors, including microvascular invasion (MVI) and poorly differentiated HCC, are also well-established risk factors for HCC recurrence following curative treatments [8,17,18,19]. Moreover, the response to local regional therapy is also an effective marker for predicting outcomes after LT [20,21].

Recent years have seen significant advances in the treatment of unresectable HCC with the emergence of immune checkpoint blockers (ICBs) [22]. ICBs targeting programmed cell death protein 1 (PD-1), programmed cell death ligand 1 (PD-L1) or cytotoxic T-lymphocyte-associated antigen 4 (CTLA-4) have recently undergone several clinical trials and are paving the way for the evolution of novel treatment strategies of advanced HCC [23]. Moreover, recent studies have demonstrated the importance of immunological markers, including PD-1 and PD-L1, and tumor microenvironments (TMEs) such as regulatory T cells (Tregs) and cytotoxic T cells (CTLs), in the recurrence of HCC following curative treatments [24,25,26]. Indeed, based on this rationale, several randomized controlled trials are evaluating the effectiveness of ICBs targeting PD-(L)1 and CTLA-4 on HCC recurrence after curative treatments [24,25].

In this review, we discuss the immunological markers and prognostic factors associated with the recurrence of HCC after curative treatments, including LT, resection, and percutaneous ablation. Moreover, we attempted to describe current challenges in the treatment of HCC following curative treatments to provide future directions for HCC treatments (Figure 1).

## 2. Liver Transplantation (LT)

### 2.1. Clinical Outcomes and Biomarkers for Recurrence

LT provides excellent long-term survival, and the expected five-year survival rate of LT in selected patients according to the Milan criteria (one tumor ≤ 5 cm or three or fewer tumors with all ≤3 cm without vascular invasion or metastasis) is 70–80% [15]. Several criteria, such as the UCSF criteria (one tumor ≤ 6.5 cm or 2–3 tumors ≤ 4.5 cm with a total of less than 8 cm) and the Upto-7 criteria (the sum of the size of the largest tumor (cm) and the number of HCC ≤ 7), have extended the range of recipient criteria, showing only slightly lower survival than the Milan criteria [27,28,29].

The level of alpha-fetoprotein (AFP) is a reliable biomarker for the recurrence of HCC patients after LT [30]. Indeed, a high AFP level greater than 1000 ng/mL before LT shows particularly poor survival after LT [31], and the French AFP Model, the Metro-Ticket 2.0, the Seoul criteria and the RETREAT score include the AFP levels for the selection of appropriate patients for LT [32,33,34,35]. Moreover, the reduction in AFP levels after local regional therapy before LT is also significantly associated with the recurrence of HCC after LT [36,37].

Aside from AFP, *Lens culinaris* agglutinin-reactive alpha-fetoprotein (AFP-L3) and des-gamma-carboxyprothrombin (DCP—also known as protein induced by vitamin K absence or antagonist-II, PIVKA-II)—are also well-known biomarkers for the recurrence of HCC after LT [38,39]. AFP-L3, a fucosylated variant of AFP, is thought to be more specific than AFP [40], and its percentage over total AFP and absolute value might be used as a biomarker for the diagnosis and recurrence of HCC after LT [38,41]. DCP has been identified in the plasma of HCC patients and is associated with portal vein invasion and poorly differentiated HCC [42,43,44]. Moreover, a high level of DCP before LT also has predictive power for recurrence after LT [38,39,45].

The uptake of ^18^F-fluorodeoxyglucose (^18^F-FDG) at positron emission tomography/computed tomography (PET-CT) is also suggested as a good biomarker for the prediction of HCC recurrence after LT [46,47]. ^18^F-FDG-avid tumors are correlated with poorly differentiated tumors and MVI, and several studies documented the combined usage of ^18^F-FDG PET-CT and clinical criteria, including the UCSF criteria and the Milan criteria, for the selection of eligible HCC patients for LT [48,49,50].

### 2.2. Immunological Markers for the Prediction of HCC Recurrence

There have been several immunological markers predicting HCC recurrence after LT. First, C-reactive protein (CRP) is an acute-phase protein synthesized by hepatocytes following systemic inflammation, that is regulated by proinflammatory cytokines such as IL-6 [51] and has been analyzed for the prediction of recurrence after LT. However, its role as a risk factor for HCC recurrence after LT is still controversial; a high CRP level was an independent factor for HCC recurrence in patients beyond the Milan criteria in one study, whereas another study showed no significance for recurrence after LT [52,53].

The neutrophil-to-lymphocyte ratio (NLR) is another inflammatory and immunologic marker that has been evaluated as a predictor for post-LT HCC recurrence. The elevation of the NLR may be related to an increase in neutrophils, a consequence of the inflammatory response, a decrease in lymphocytes, and a lower immunological suppression of tumor growth [54]. Several studies have documented that a high NLR with a cutoff value from 3 to 5 is a significant risk factor for HCC recurrence after LT [55,56,57]. Recently, the MORAL score, including NLR as one of pre-MORAL factors, has been developed and provides high accuracy for recurrence after LT [58].

Interestingly, in patients with a high NLR, peritumoral IL-17 levels and CD163-positive tumor-associated macrophages (TAMs) were significantly increased [55]. In HCC patients, TAMs, interacting with IL-17-producing cells, have been found to promote tumor proliferation, and CD163-positive TAMs play a suppressive role in the antitumor immune response, which may impact the recurrence of HCC [59,60]. Although further studies are needed to confirm the ability to predict HCC recurrence, peritumoral IL-17 levels and TAMs, including CD163-positive TAMs, may be used as predictive markers for post-LT HCC recurrence.

Furthermore, the level of plasma vascular endothelial growth factor (VEGF), a protein promoting angiogenesis, may have a prognostic role in post-LT HCC recurrence [61,62]. A high level of VEGF was associated with a higher recurrence rate after LT [61], and a recent study revealed the prognostic role of VEGF in association with the Milan criteria. Higher VEGF levels were associated with poor prognosis in LT patients within the Milan criteria, whereas lower VEGF-C levels correlated with better outcomes in patients exceeding the Milan criteria [62]. Considering the significant role of VEGF in the angiogenesis of tumors and as a treatment target in advanced HCC, these biomarkers need to be validated in the setting of further large-scaled studies.

The type and amount of lymphocyte infiltration to the tumor can also affect the prognosis of LT. The CD4:CD8 ratio was identified as an independent predictor of recurrence [63]. However, the infiltration of CD4^+^CD25^+^Foxp3^+^ Tregs was not associated with recurrence but correlated with vascular invasion and poor tumor differentiation tumor [63,64]. A high CD4^+^ T cells over CD8^+^ T cells imply the higher portion of HCC-specific CD4^+^ T cells, functional T cells against tumor [63,65], whereas the infiltrated Tregs functionally inhibit anti-tumor response, which can lead to poor prognosis after LT (Table 1) [63,66]. Further studies with large number of patients are necessary to confirm the impact of infiltrated lymphocytes on the recurrence of HCC after LT.

Meanwhile, there have been efforts to discover risk factors and immunological mechanisms for HCC recurrence following downstaging of HCC before LT. Along with baseline tumor burden, poor treatment response to TACE before LT has also been identified as a risk factor for HCC recurrence following LT [67,68]. Notably, patients with high expression of PD-L1 at baseline was associated with not only poor treatment response but also poor survival after TACE [69]. Moreover, the reduction of Tregs after TACE were associated with improved outcomes [70]. Based on these results, in patients who receive LT after downstaging of HCC, the PD-1-PD-L1 axis and Tregs affect HCC recurrence rates but further confirmatory studies are required.

### 2.3. Challenges in HCC Treatment following Recurrence after LT

One of the distinct features of LT patients is that they must take life-long immunosuppressants (ISs), which may affect HCC recurrence and treatment outcomes. High levels of calcineurin inhibitors (CNIs), such as tacrolimus and cyclosporine, could possibly contribute to the recurrence of HCC after LT [71,72,73]. Meanwhile, mammalian target of rapamycin (mTOR) inhibitors, including sirolimus and everolimus, may decrease HCC recurrence after LT [74,75,76]. mTOR inhibitors target the phosphatidylinositol 3-kinase (PI3K)/Akt/mTOR signaling pathway, which has at least two mTOR complexes, m-TOR complex 1 and m-TOR complex 2, that can lead to HCC proliferation and spreading [77,78]. Considering anticancer mechanism of mTOR inhibitors, the addition of mTOR inhibitors with a reduction in CNI dosage may decrease HCC recurrence after LT [79,80]. However, because the reduction in HCC recurrence following the use of mTOR inhibitors has only been demonstrated in relatively small studies, high-quality studies, including well-designed multicenter studies, are needed to confirm the cancer preventive effects of mTOR inhibitors.

Another important issue in the treatment of HCC after LT is the risk of rejection. Sorafenib treatment for recurrent HCC after LT has been shown to be safe and effective, and demonstrated better outcomes compared to advanced HCC without LT [81,82]. However, there is a concern for the use of ICBs targeting PD-1/PDL-1 or CTLA-4 in LT patients with recurrent HCC. The PD-1/PDL-1 pathway in organ transplantation plays a pivotal role in the regulation of alloimmunity and the establishment of graft tolerance [83,84,85], and the usage of ICBs can increase the risk of rejection in LT patients. Overall, approximately 25–50% of transplanted patients experienced graft rejection, and some patients had end-stage organ failure even after salvage IS treatment [86,87,88,89]. Thus, it is not currently recommended to use ICBs in LT patients because of the high risk of rejection, which could be even life-threatening for some LT patients [90]. Future researches are necessary to classify and select eligible LT patients for ICB treatment to reduce the risk of rejection.

## 3. Surgical Resection

### 3.1. Clinical Outcomes and Biomarkers for Recurrence

Surgical resection also provides good outcomes, and in well-selected patients, the five-year survival rate is 50–80% [91]. Generally, patients with BCLC stage 0 or A disease are eligible for resection, and the liver function along with the extent of resection are the two most important factors to determine the feasibility of resection [8,92]. In noncirrhotic patients, liver resection is the treatment of choice with a low rate of postoperative liver decompensation [93] while major hepatectomy and resection of three or more Couinaud segments, should be considered in patients without portal hypertension and with a model for end-stage liver disease (MELD) score < 9 [94]. Although the presence of portal hypertension was previously considered a contraindication for resection, recent studies demonstrated that resection can be performed despite the presence of portal hypertension (platelet count < 100,000/mL and/or splenomegaly), as long as MELD score is less than 9 points [6].

Meanwhile, the 5-year recurrence rate after resection has been reported to be approximately 40–70% [11,95,96] and, therefore, finding high-risk patients for recurrence is a very important issue. Traditionally, large tumors (>5 cm), multiple lesions or satellite nodules are well-known risk factors for recurrence after resection [97,98,99]. Histopathological findings, including microvascular invasion and poor tumor differentiation, are also associated with HCC recurrence [97,99]. Apart from tumor and histological parameters, several biomarkers have also been evaluated for predicting recurrence after resection. To date, the level of AFP is the most evaluated biomarker for recurrence after resection. Although the exact cutoff level has not been determined, a high AFP level is strongly associated with HCC recurrence [100,101]. Recently, with the inclusion of AFP levels, pre- and postoperative models have been developed to predict HCC recurrence after resection using a large international cohort [101]. The AFP-L3 and DCP levels may also be potential biomarkers for postresection HCC recurrence [102,103].

By analyzing the histopathology of resected tumors, several novel biomarkers, including stem cell markers, have been evaluated for predicting HCC recurrence. Positivity for epithelial cell adhesion molecule (EpCAM) and cytokeratin-19 (CK-19) which are markers of liver cancer stem cells expressed in hepatic progenitor cells have been identified as poor prognostic factors for recurrence [104,105]. Glypican-3, an oncofetal protein, also has been shown to predict HCC recurrence [106]. Further studies to evaluate and validate such biomarkers in the setting of recurrence are needed.

In addition, a novel concept of liquid biopsy and its potential to diagnose and predict prognosis of HCC is currently under research [107]. Using liquid biopsy performed at the portal vein or hepatic vein, the clinical applications of circulating tumor cells (CTCs) and cell-free DNA (cfDNA), which are crucial components of the liquid biopsy, have been evaluated [108]. CTC is usually defined as nucleated cells expressing EpCAM and CK-8, 18, and/or 19, while being negative for CD45, leukocyte-specific antigen [109], and the preoperative presence of CTCs was significantly associated with HCC recurrence after resection [107,110,111]. Moreover, the amount, the presence of methylation, and the mutations of cfDNA were also associated with recurrence of HCC [107,112]. These results may pave the way to the development of novel, reliable and less invasive biomarkers of HCC recurrence in the future.

### 3.2. Immunological Markers for the Prediction of HCC Recurrence

The prognostic role of CRP in HCC recurrence after resection has been evaluated, and a high level of CRP has been shown to be independently associated with recurrence in several studies [113,114,115]. The level of CRP, reflecting inflammation, is regulated by interleukin (IL)-6 and IL-1β, which are associated with carcinogenesis, angiogenesis, and tumor growth [51,116]. Although the underlying mechanism is unclear, such relation may explain the association between CRP and recurrence after resection. The NLR, TAM levels, inflammatory and immunologic markers have also been associated with recurrence after resection [117,118,119,120]. A high NLR (cutoff value 2–3) was an independent prognostic factor for recurrence and has been associated with increased infiltration of TAMs [118,121]. TAMs express IL-6 and IL-8, which promote systemic neutrophilia, and CD163-positive TAMs also produce IL-10, which can suppress antitumor immune response, resulting in an increase in the risk of recurrence (Table 1) [122,123,124].

The other potential immunologic markers for predicting recurrence are the levels of VEGF(R1) and PD-(L)1. Different from the prognostic ability of VEGF in LT patients, the role of VEGF level is controversial in patients who receive resection. One study demonstrated that a high serum level of VEGF, a potent stimulator of angiogenesis [125], was associated with recurrence [126]. However, another study showed that plasma VEGF was not a risk factor for recurrence, and only a high level of VEGFR1 was a significant factor [62]. Considering the role of the VEGF pathway in tumor growth and angiogenesis [125], further detailed studies evaluating the prognostic role of VEGF in patients with resection are needed. The peritumoral and circulating PD-L1 in patients who underwent hepatic resection have been also examined and documented as a significant factor for recurrence in various studies [127,128,129,130]. PD-L1 is one of the PD-1 ligands, and its ligation to PD-1 can lead to exhaustion and apoptosis of T-cells, which causes immune suppression leading to tumor growth and metastasis [131,132]. These findings provide insights into the importance of PD-(L)1 expression in the outcomes of HCC patients after resection.

Several studies also evaluated the impact of intra- and peritumoral immune cells, such as Tregs, CTLs, myeloid-derived suppressor cells (MDSCs), hepatic stellate cells (HSCs) and dendritic cells (DCs), on the recurrence of HCC after resection. A high density of intratumoral Tregs in combination with low CD4^+^ or CD8^+^ CTLs, suggesting immune suppression against tumors, were independent risk factors for recurrence after resection [64,133,134,135]. Reciprocally, a high density of CD3^+^, CD4^+^, CD8^+^ T cells in the intra- and peritumor was associated with a markedly reduced rate of HCC recurrence by generating the anticancer immune response [135,136]. An increased frequency of circulating MDSCs, which are known to suppress the host immune system, also correlated with early recurrence after resection [137]. Moreover, a high density of peritumoral HSCs, mesenchymal cells enhancing inflammation and fibrosis, intratumoral DCs, and antigen-presenting cells regulating adaptive immunity, were also indicators for poor prognosis of HCC following resection [138,139]. These findings suggest that intra- and peritumoral immune cells may represent as prognostic markers as well as therapeutic targets for HCC treatment. Furthermore, one study reported that the combination of elevated EpCAM CTCs and Tregs may be an indicator for early HCC recurrence following resection [112,140]. Therefore, the development of models using both clinical factors and immunological factors to identify high-risk patients for recurrence after resection is needed in the future.

### 3.3. Challenges in Prevention of HCC Recurrence

Because of the high rate of recurrence after resection, there has been a need for adjuvant treatment after resection. However, no adjuvant treatment is currently recommended in major treatment guidelines, with negative results in various clinical trials and studies [6,8]. A phase III randomized controlled trial, evaluating the efficacy of sorafenib versus placebo as an adjuvant therapy in HCC patients after curative treatments failed to document any positive results [141], although the selection bias and side effects of sorafenib may have affected the results of the study.

Recently, combination therapy with a PD-L1 inhibitor (atezolizumab) and VEGF inhibitor (bevacizumab) demonstrated remarkable outcomes compared to sorafenib in the treatment of unresectable HCC [22]. These results prove that anti-VEGF treatment enhances the efficacy of PD-L1 inhibition in HCC treatment. As we mentioned above, intra- and peritumoral immunosuppressive environments are associated with poor outcomes, and VEGF can also exert immunosuppressive effects by downregulating T-cell activation, reducing T-cell infiltration, and increasing Tregs and MDSCs [142,143,144]. Along with Tregs and MDSCs, PD-1 causes immunosuppression in HCC [145], and these data may provide a rationale for the adjuvant usage of combination therapy, including PD-L1/VEGF blockade, in addition to curative therapies. Moreover, the combined analysis of CTCs and immune cells and their interaction might facilitate the therapeutic decision-making in HCC.

Currently, a number of studies are underway, evaluating various drugs and immunotherapies in neoadjuvant and/or adjuvant settings after resection [92,146]. Most studies include high-risk patients for recurrence, such as multiple tumors, large single tumors (>3 cm), and tumors with MVI, and recurrence-free survival is the primary endpoint of in these trials. Drugs in these settings target the CTLA-4, PD-1 (nivolumab, pembrolizumab, cemiplimab), or PD-L1 (durvalumab, atezolizumab) signaling pathways and other checkpoint proteins [92]. In the future, these adjuvant approaches after resection may improve patient outcomes.

## 4. Percutaneous Ablation Therapy

### 4.1. Clinical Outcomes and Biomarkers for Recurrence

Percutaneous ablation therapies, including radiofrequency ablation (RFA), microwave ablation (MWA) and cryoablation are also mainstay of curative treatment of HCC. Among them, RFA is the most validated ablation therapy and recommended for patients with BCLC stage 0 and A HCC, especially for patients unsuitable for surgery [6,8]. The rate of complete ablation in tumors ≤ 5 cm is over 95% [147,148,149], and RFA shows comparable outcomes with resection in the treatment of a single tumor ≤ 3 cm [150]. Moreover, RFA is the most cost-effective treatment in BCLC 0 tumors and two or three nodules ≤ 3 cm [151].

After successful RFA treatment, the local tumor recurrence rate is 10–30%, whereas distant recurrence at 5 and 10 years are 58–81% and 80–88%, respectively [152]. Two major risk factors for local recurrence are tumor size (>3 cm) and the presence of tumor-adjacent vessels [153,154]. Meanwhile, distant recurrence is related to de novo carcinogenesis and multiple tumor factors, such as tumor size, number of tumors, and AFP level [155,156]. The model to predict tumor recurrence after LDLT (MoRAL) score using AFP and PIVKA-II might also be used to evaluate high-risk patients for tumor recurrence after RFA [157].

Moreover, the differentiation of tumor and tissue biomarkers could predict tumor recurrence after RFA. Tumors with poor differentiation as well as positive endocan, a marker of endothelial activation and MVI, are also at high risk for recurrence after RFA [158,159]. Moreover, similar to resected patients, RFA-treated patients with positive CK-19^+^ tumors were at high risk for recurrence [160]. Further research to develop a model for recurrence combining tumor characteristics, biomarkers and tumor markers is warranted.

### 4.2. Immunological Markers for the Prediction of HCC Recurrence

Ablation therapy, especially RFA, induces a variety of immunologic effects after treatment. Thermally induced necrosis induces a tumor-specific immune response, triggering inflammatory cytokines around the necrotic zone and increase in cytotoxic T-cells (CTLs) [161,162,163,164]. Similar to RFA, the number of peripheral T cells and the ratio of Th1/Th2 cytokines are increased after MWA treatment [165]. These activated T cells may be associated with activation of DCs, which causes the increase in tumor necrosis factor-α and IL-1β [166]. Indeed, the number of tumor-associated antigen (TAA)-specific T cells after RFA was a predictive factor for the prevention of HCC recurrence [167]. However, the number of TAA-specific T cells, which was also inversely correlated with MDSCs, were decreased 24 weeks after RFA, which could be insufficient to prevent recurrence in the long term [167]. Along with cytotoxic T-cells, the frequency of MDSCs was also associated with HCC recurrence and prognosis in HCC patients after RFA [168]. Therefore, upregulation of TAA-specific CTLs with the inhibition of MDSCs may improve the prognosis of RFA-treated patients.

A high serum level of VEGF, a potent angiogenic factor, was also associated with a greater risk of recurrence after RFA [169]. Immunosuppressive microenvironments, such as reduction in T-cell infiltration and increase in Tregs and MDSCs, are also thought to be modulated by VEGF [142,143,144]. Moreover, insufficient RFA can promote angiogenesis and residual hepatocellular cell migration via hypoxia inducible factor-1 alpha (HIF-1α)/VEGFA signaling [170,171]. HIF-1α is a pivotal regulator of the adaptive response to hypoxia, which is highly expressed in hypoxic conditions and contributes to angiogenesis and metastasis, which can lead to poor prognosis of HCC patients [172,173]. Furthermore, the increase in circulating PD-L1/PD-1 expression and Th17 cells after cryoablation and MWA were also associated with tumor recurrence [174,175]. Taken together, the evidence suggested so far support future studies targeting immunosuppressive microenvironments, immune checkpoints, and VEGF to reduce the recurrence of HCC after RFA (Table 1).

### 4.3. Challenges in Prevention of HCC Recurrence

Because of the high rate of recurrence after RFA, there has been an unmet need for adjuvant treatment after RFA. As we mentioned above, ablation therapy induces not only tumor necrosis but also immunosuppressive microenvironments, including reduced TAA-specific CTLs and increased Tregs and MDSCs via the HIF-1α/VEGF signaling pathway. Moreover, the PD-1/PD-L1 and CTLA-4 signaling pathways can cause deactivation of TAA-specific CTLs, and intratumoral PD-L1 expression is associated with tumor aggressiveness [129,176]. Meanwhile, TAA-specific T cell response after MWA was associated with longer progression-free survival [177]. Based on this rationale, ICBs targeting the PD-1/PD-L1, CTLA-4, and VEGF pathways may strengthen the immune response against possible residual tumors after RFA.

Currently, several phase II or III trials are evaluating anti-PD-1/L-1 inhibitors as single agents or in combination with CTLA-4 inhibitors, VEGF inhibitors or tyrosine kinase inhibitors (TKIs) as adjuvant therapies after curative treatments, including RFA [178,179]. Although the STORM trial evaluating sorafenib in the adjuvant setting failed to show superior recurrence-free survival compared to a placebo [141], promising developments made in ICBs may lead us to a new era for the treatment of HCC by reducing recurrence after curative therapies, including RFA.

## 5. Conclusions

Along with tumor-related factors, including tumor size, number, and differentiation, several markers, including AFP, AFP-L3, DCP, CRP and NLR, were also associated with recurrence after curative therapies. Moreover, immunological markers and TMEs are also important factors for recurrence. The increase in VEGF, PD-1/L-1, TAMs and immune suppressive cells, including Tregs and MDSCs, correlated with HCC recurrence, whereas the decrease in CTLs was associated with recurrence of HCC after curative treatments. Therefore, there have been attempts to improve TMEs by increasing CTLs and decreasing immune suppressive cells.

In LT patients, using mTOR inhibitors and, targeting the PI3K/Akt/mTOR pathway, may reduce HCC recurrence although validation should be made in large-scale studies to prove its effectiveness. Furthermore, due to the risk of rejection after ICB treatment in some LT patients with recurrent HCC, it may be necessary to find and classify eligible patients for ICB in the future. In HCC patients treated with curative resection and ablation, there are many ongoing trials using ICBs, targeting PD-(L)1, VEGF, and CTLA-4, in adjuvant settings after curative therapies. The results of these ongoing adjuvant studies using ICBs may not only change the treatment landscape of early HCC but may also pave the way to a new era of adjuvant therapy after curative treatments.

## Figures and Tables

**Figure 1 ijms-22-10271-f001:**
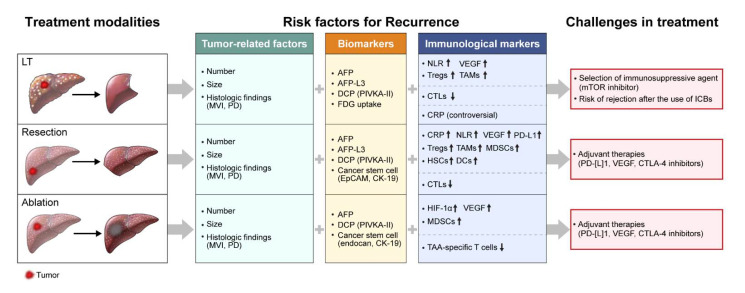
Risk factors for recurrence and challenges following curative treatments for hepatocellular carcinoma. LT, liver transplantation; MVI, microvascular invasion; PD, poor differentiation; AFP, alpha-fetoprotein; AFP-L3, Lens culinaris agglutinin-reactive alpha-fetoprotein; DCP, des-gamma-carboxyprothrombin; PVIKA-II, protein induced by vitamin K absence or antagonist-II; FDG, fluorodeoxyglucose; EpCAM, epithelial cell adhesion molecule; CK-19, cytokeratin-19; NLR, neutrophil-to-lymphocyte ratio; VEGF, vascular endothelial growth factor; Treg, regulatory T cell; TAM, tumor-associated macrophage; CTL, cytotoxic T-cell; CRP, C-reactive protein; PD-L1, programmed cell death ligand 1; MDSC, myeloid-derived suppressor cell; HSC, hepatic stellate cell; DC, dendritic cell; HIF-1α, hypoxia inducible factor-1 alpha; TAA, tumor-associated antigen; mTOR, mammalian target of rapamycin; ICB, immune checkpoint blocker; PD-1, programmed cell death protein 1; CTLA-4, cytotoxic T-lymphocyte-associated antigen 4.

**Table 1 ijms-22-10271-t001:** Immunological factors for recurrence of hepatocellular carcinoma following curative treatments.

Immune Cells/Signaling Pathway	Findings in HCC Recurrence
Cytotoxic T cells	A decrease in infiltrated CD4^+^, CD8^+^ T cell associated with recurrence ↑
Immunosuppressive cells	
Tregs	An increase in Tregs correlated with recurrence ↑
MDSCs	An increase in MDSCs associated with recurrence ↑
Macrophages	An increase in tumor-associated macrophage associated with recurrence ↑
VEGF	An increase in VEGF associated with recurrence ↑
PD-1/PD-L1	An increase in PD-L1 correlated with high-risk factors for recurrence

Tregs, regulatory T cells, MDSCs, Myeloid derived suppressor cells; VEGF, vascular endothelial growth factor; PD-1, programmed cell death protein 1; PD-L1, programmed cell death ligand 1.

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
