# Peer review of "Immunological Markers, Prognostic Factors and Challenges Following Curative Treatments for Hepatocellular Carcinoma"

_ijms, 2021, doi:10.3390/ijms221910271_

Round 1

Reviewer 1 Report

The paper is well written, well organized and it is ready to accept for publication with minor English editing (style).

Author Response

Ans) The authors would like to thank the comment of the reviewer and fully agree with the opinions of the reviewer. Following the reviewer’s comment, we edited the style of English in our revised manuscript. We hope that our explanations and revisions are satisfactory.

Reviewer 2 Report

Very interesting and well written.

I think it should be interesting to add something on liquid biopsy and circulating tumor cells and on the impact of downstaging treatment on adverse prognostic factors

Author Response

Ans) The authors appreciate the comment of the reviewer and totally agree with the opinions of the reviewer. Following the reviewer’s comment, we added the concept and impact of liquid biopsy, circulating tumor cells (CTCs), and cell-free DNA (cfDNA) on HCC recurrence in our revised manuscript. Indeed, the presence of CTCs and the amount of cfDNA were associated with HCC recurrence following surgical resection. Moreover, the combination of elevated CTCs and Tregs might be an indicator of early recurrence of HCC. We also added the possible prognostic factors for HCC recurrence in patients with downstaging of HCC before LT. The level of PD-L1 and Tregs were associated with poor response after TACE, which is one of risk factors for HCC recurrence after LT. Therefore, the PD-1-PD-L1 axis and Tregs might also affect HCC recurrence after LT.

We hope that our explanations and revisions are satisfactory, and we made revisions to the manuscript as follows:

“Meanwhile, there have been efforts to discover risk factors and immunological mechanisms for HCC recurrence following downstaging of HCC before LT. Along with baseline tumor burden, poor treatment response to TACE before LT has also been identified as a risk factor for HCC recurrence following LT [67,68]. Notably, patients with high expression of PD-L1 at baseline was associated with not only poor treatment response but also poor survival after TACE [69]. Moreover, the reduction of Tregs after TACE were associated with improved outcomes [70]. Based on these results, in patients who receive LT after downstaging of HCC, the PD-1-PD-L1 axis and Tregs affect HCC recurrence rates but further confirmatory studies are required.” (On page 4)

“In addition, a novel concept of liquid biopsy and its potential to diagnose and predict prognosis of HCC is currently under research [107]. Using liquid biopsy performed at portal vein or hepatic vein, the clinical applications of circulating tumor cells (CTCs) and cell-free DNA (cfDNA), which are crucial components of the liquid biopsy, have been evaluated [108]. CTC is usually defined as nucleated cells expressing EpCAM and CK-8, 18, and/or 19, while being negative for CD45, leukocyte-specific antigen [109], and the preoperative presence of CTCs was significantly associated with HCC recurrence after resection [107,110,111]. Moreover, the amount, the presence of methylation, and the mutations of cfDNA were also associated with recurrence of HCC [107,112]. These results may pave the way to the development of novel, reliable and less invasive biomarkers of HCC recurrence in the future.” (On page 6)

“Furthermore, one study reported that the combination of elevated EpCAM CTCs and Tregs may be an indicator for early HCC recurrence following resection [112,140]. Therefore, the development of models using both clinical factors and immunological factors to identify high-risk patients for recurrence after resection is needed in the future.” (On page 7)

“Along with Tregs and MDSCs, PD-1 causes immunosuppression in HCC [145], and these data may provide a rationale for the adjuvant usage of combination therapy, including PD-L1/VEGF blockade, in addition to curative therapies. Moreover, the combined analysis of CTCs and immune cells and their interaction might facilitate the therapeutic decision-making in HCC.” (On page 7)
